# Diffuse Myocardial Fibrosis on Cardiac Magnetic Resonance Imaging Is Related to Galectin-3 and Predicts Outcome in Heart Failure

**DOI:** 10.3390/biom13030410

**Published:** 2023-02-22

**Authors:** Elles M. Screever, Thomas M. Gorter, Tineke P. Willems, Joseph Pierre Aboumsallem, Navin Suthahar, Belend Mahmoud, Dirk J. van Veldhuisen, Rudolf A. de Boer, Wouter C. Meijers

**Affiliations:** 1Department of Cardiology, University Medical Center Groningen, P.O. Box 30.001, 9700 RB Groningen, The Netherlands; 2Department of Cardiology, Thorax Center, Erasmus University Medical Center, P.O. Box 2040, 3000 CA Rotterdam, The Netherlands; 3Department of Radiology, University Medical Center Groningen, 9700 RB Groningen, The Netherlands

**Keywords:** heart failure, cardiac magnetic resonance imaging, post-contrast T1 time, focal fibrosis, diffuse fibrosis, galectin-3

## Abstract

Aims: Ongoing adverse remodeling is a hallmark of heart failure (HF), which might be reflected by either focal or diffuse myocardial fibrosis. Therefore, in (pre)clinical settings, we used immunohistochemistry or cardiac magnetic resonance imaging (CMR) to investigate the association of (focal or diffuse) fibrosis with cardiac biomarkers and adverse events in HF. Methods and results: In C57Bl/6J mice, we determined the presence and extent of myocardial fibrosis 6 weeks post-myocardial infarction (MI). Furthermore, we studied 159 outpatient HF patients who underwent CMR, and determined focal and diffuse fibrosis by late gadolinium enhancement (LGE) and post-contrast T1 time of the non-LGE myocardium, respectively. HF patients were categorized based on the presence of LGE, and by the median post-contrast T1 time. Kaplan–Meier and Cox regression analyses were used to determine the association of fibrosis with HF hospitalization and all-cause mortality. LGE was detected in 61 (38%) patients. Cardiac biomarker levels were comparable between LGE-positive and LGE-negative patients. LGE-positive patients with a short T1 time had elevated levels of both NT-proBNP and galectin-3 (1611 vs. 453 ng/L, *p* = 0.026 and 20 vs. 15 μg/L, *p* = 0.004, respectively). This was not observed in LGE-negative patients. Furthermore, a short T1 time in LGE-positive patients was associated with a higher risk of adverse events (log-rank *p* = 0.01). Conclusion: This study implies that cardiac biomarkers reflect active remodeling of the non-infarcted myocardium of patients with focal myocardial scarring. Diffuse fibrosis, in contrast to focal scarring, might have a higher prognostic value regarding adverse outcomes in HF patients.

## 1. Introduction

Following acute myocardial infarction (MI), a cascade of cardiac immune cell infiltration, cardiomyocyte necroptosis and formation of granulation tissue contributes to the formation of a focal myocardial scar [1]. Although the presence and extent of myocardial scarring have shown prognostic value [2,3,4,5], focal scarring is also considered essential to maintain ventricle wall integrity. Interestingly, on the contra-lateral side, diffuse or interstitial fibrosis plays an important role in the onset and progression of cardiovascular disease, and mainly results in decreased cardiac function [6]. In fact, this type of fibrosis might potentially be more interesting because it is, at least in part, reversible.

Cardiac magnetic resonance (CMR) imaging has evolved as the gold standard noninvasive imaging technique for myocardial tissue characterization. Late gadolinium enhancement (LGE) and T1 mapping techniques allow the determination and quantification of focal myocardial scar formation and detailed tissue characterization including diffuse fibrosis, respectively [7].

In this study, we determined the presence of myocardial fibrosis—either focal myocardial scarring or diffuse fibrosis—and evaluated its relation with cardiac biomarkers and outcomes. For this purpose, we used an MI murine model and a well-characterized cohort of outpatient heart failure (HF) patients, either using immunohistochemistry methods or CMR metrics (i.e., LGE and post-contrast T1 time), respectively.

## 2. Methods

### 2.1. Animal Studies

#### 2.1.1. Mouse Model

MI was induced in mice with a C57Bl6/J background (The Jackson Laboratory, Bar Harbor, ME) by permanent ligation of the left anterior descending coronary artery (LAD). In sham-operated animals, the suture was placed under the artery and removed without ligation. After a follow-up of 6 weeks, mice were sacrificed [8]. In total, analyses were performed on 20 MI mice and 9 sham-operated mice.

#### 2.1.2. Immunohistochemistry

Masson’s trichrome staining was performed to determine focal and diffuse myocardial fibrosis in 4 µm paraffin-embedded cardiac tissue sections. Whole stained sections were scanned with a high-throughput scanning system (Hamamatsu, Japan). Focal fibrosis was described as dense and boundless fibrotic patterns replacing the original myocardial tissue structure, as visually detected by an independent researcher. Diffuse fibrosis of the non-infarcted myocardium was defined as interstitial fibrosis evenly distributed between cardiomyocytes, with relatively well-organized myocardial bundles and intact cardiomyocyte morphology or following anatomical structures. Diffuse fibrosis was determined by analyzing all cardiac tissue without evident focal fibrosis. The amount of diffuse fibrosis was quantified as the percentage of positive cells compared to the total scanned area (Aperio ImageScope v12.4.3.5008) and represented as fold change.

#### 2.1.3. Quantitative Polymerase Chain Reaction

Total RNA was isolated from snap-frozen left ventricles (LV) using TRI reagent (Sigma-Aldrich, St Louis, MO, USA). cDNA synthesis was performed using the QuantiTect Reverse Transcription Kit (Qiagen, Germany). A real-time polymerase chain reaction (PCR) was carried out using SYBR Green (Bio-Rad, Hercules, CA, USA) using the Bio-Rad CFX384 Real-Time PCR System with associated software (Bio-Rad, Hercules, CA, USA). Gene expression was normalized with the mean of 36B4 mRNA content.

### 2.2. Human Studies

#### 2.2.1. Patient Cohort

Consecutive outpatient HF patients who visited the outpatient clinic of the University Medical Center Groningen (UMCG), in Groningen, The Netherlands, between March 2014 and December 2017 were included in this study. Their data were published previously [9]. In total, 842 patients were included in this study cohort, of which 159 patients (19%) underwent CMR with administration of gadolinium and were included in present analyses. The primary endpoint was defined as a composite of HF hospitalization and all-cause mortality, first to occur, after a median follow-up time of 2.6 [2.1–3.1] years. HF hospitalization was defined as an unplanned overnight stay in the hospital due to acute cardiac decompensation.

All patients were ≥18 years of age and were treated according to the European Society of Cardiology (ESC) guidelines [10].

#### 2.2.2. CMR Protocol

CMR examinations were performed on a 1.5-T scanner (Siemens Aera, Erlangen, Germany). ECG-gated cine true fast imaging with steady-state free precession (True FISP) sequences with breath holding were performed in three long-axis slices and in contiguous short-axis slices covering the entire LV. The following scan parameters were used: TE 1.1 ms, TR 42 ms, flip angle 55°; matrix 192 × 192, voxel size 1.82 × 1.82 × 8 mm; slice thickness 8 mm; and slice gap 2 mm. Using the short-axis slices, the endo and epicardial borders of the LV were semi-automatically traced and manually adjusted by experienced laboratory technicians using the available software (QMass 7.6, Medis, Leiden, The Netherlands). The LV end-diastolic volume (EDV), end-systolic volume (ESV), stroke volume (SV), ejection fraction (EF) and mass were calculated using the summation of slices multiplied by slice thickness method.

#### 2.2.3. LGE Analysis

Using identical slice location, LGE images were acquired 10 min after intravenous administration of 0.2 mmol/kg gadolinium-based contrast agent (Dotarem, Gorinchem, The Netherlands) with a single shot 2D phase sensitive inversion recovery sequence (TE 3.2 ms, TR 700 ms; flip angle 25°; matrix 360 × 360 mm, voxel size 1.4 × 1.4 × 8 mm; slice thickness 8 mm, slice gap 2 mm) to identify the location and extent of focal late enhancement. The inversion time was individually set to null the signal of the normal myocardium. The presence of focal LGE was visually determined by one observer (T.M.G.) and reviewed by another observer (T.P.W.). LGE size was quantified with the full width at half maximum technique using QMass 7.6 (Medis, Leiden, The Netherlands), and was expressed as a percentage of the total LV mass. The LGE pattern was classified as ischemic (e.g., subendocardial or transmural pattern) and non-ischemic (e.g., mid-wall, epicardial or global pattern) based on previous recommendation [11].

#### 2.2.4. T1 Measurements

Post-contrast T1 measurements were performed using an inversion recovery Look-Locker sequence (True FISP), similar to the T1 scout sequence (TE 1.1 ms; TR 23 ms; flip angle 30°; matrix 380 × 380 mm, voxel size 1.98 × 1.98 × 8 mm). The Look-Locker sequence was performed 10 min after the administration of 0.2 mmol/kg gadolinium in a short-axis slice at mid-level over two RR intervals with varying inversion times. The endo- and epicardial borders of the LV were manually traced and adjusted for each image by one observer (T.M.G.) using QMass 7.6 (Medis, Leiden, The Netherlands). Special attention was paid to exclude the blood volume. By drawing regions of interest (ROIs) within the endo- and epicardial borders of the LV, and by excluding regions with focal LGE, the post-contrast T1 time of the non-LGE myocardium was determined by fitting the signal intensity to an analytical expression of the T1 inversion recovery [12]:I = A − B exp(−t/T1*)(1)
T1 = T1*((B/A) − 1)(2)

In Equation (1), the signal intensity (I) is obtained from the average image intensity inside the ROI. The time (t) is the effective inversion time associated with each image. A and B are the scaling and offset constants, respectively. Together with the relaxation time (T1*), they are obtained by nonlinear minimization (fitting) using the Levenberg–Marquardt algorithm. T1 can be obtained from the T1* by performing a correction (Equation (2)) [12].

#### 2.2.5. Biochemical Measurements

Galectin-3 levels were measured in EDTA plasma using a chemiluminescent microparticle immunoassay (CMIA) on an Abbott ARCHITECT automated immunoassay analyser (Abbott Park, IL, USA). N-terminal B-type natriuretic peptide (NT-proBNP) was measured in Lithium Heparin plasma using a commercially available electrochemiluminescent sandwich immunoassay analyzed on a Roche Modular platform (Roche, Mannheim, Germany).

### 2.3. Statistics

Data are presented as means (±SD) when normally distributed and as medians (interquartile range [IQR]) when non-normally distributed. Categorical variables are presented as number (%). The differences between the two groups were analyzed with the use of the Student’s *t*-test for normally distributed data, the Mann–Whitney U test for non-normally distributed data and the Spearman’s chi square test for categorical variables. Linear regression analysis was performed to determine the association between diffuse fibrosis and outcome (i.e., LV ejection fraction, LVEF) in mice. Using Kaplan–Meier analysis and a log-rank test, we compared the incidence of the primary composite endpoint of HF hospitalization and all-cause mortality based on the presence or absence of LGE. Subsequently, LGE-positive patients were stratified based upon the median post-contrast T1 time of the non-LGE myocardium. A total of 13 LGE-positive patients were excluded prior to post-contrast T1 time analyses since T1 time measurements were not available. Univariable Cox proportional hazard models were used to evaluate the association between T1 time with the primary endpoint of HF hospitalization and all-cause mortality (per 10 milliseconds decrease in T1 time). All reported *p* values are two-tailed. A *p* value < 0.05 suggested statistical significance. All statistical analyses were conducted using STATA software version 16.0 (Stata Corp LP, College Station, TX, USA) and GraphPad Prism version 9.4.1 (GraphPad Software Inc., La Jolla, CA, USA).

## 3. Results

### 3.1. Presence of Focal Myocardial Fibrosis after MI

Some 6 weeks after surgery, Masson’s trichrome staining revealed evident myocardial scarring in the hearts of MI mice, as depicted in Figure 1A. Additionally, the hearts of MI mice showed significantly increased mRNA expression levels of genes involved in fibrosis, as depicted by a 5.0-, 5.0- and a 4.9-fold increased expression of galectin-3 (*Lgals3*, *p* = 0.0002), collagen type I alpha I chain (*Col1a1*, *p* < 0.0001) and collagen type III alpha I chain (*Col3a1*, *p* < 0.0001), respectively (Figure 1B). Additionally, increased cardiac immune infiltration was seen, as reflected by a 6.3- and 1.6-fold higher expression of interleukin-6 (*Il6*, *p* = 0.0009) and a cluster of differentiation 68 (*Cd68*, *p* = 0.0009) (Figure 1B).

### 3.2. Presence of Diffuse Fibrosis of the Non-Infarcted Myocardium

To determine the presence and extent of diffuse myocardial fibrosis after a MI, the amount of fibrosis was quantified in the non-infarcted area of the myocardium (Figure 1A). At 6 weeks post-MI, MI mice showed significantly higher levels of diffuse fibrosis compared to their sham-operated counterparts (Figure 1C, *p* < 0.0001).

To assess the relation of diffuse myocardial fibrosis with HF severity, we performed linear regression analyses between diffuse fibrosis, as assessed by Masson’s trichrome staining, and LVEF. Diffuse fibrosis was shown to be associated with LVEF (β = −0.69, *p* < 0.0001; Figure 1D).

### 3.3. LGE Positive vs. LGE Negative and Cardiac Biomarkers

A total of 159 outpatient HF patients were studied (Table 1). The mean age of the population was 59 ± 14 years, with 93 (58%) male patients. Some 126 (79%) patients had New York Heart Association (NYHA) class ≥ II symptoms, with 86 (54%) patients having a LVEF < 40%.

To study focal myocardial fibrosis and cardiac biomarkers, patients were categorized based on the presence or absence of LGE. In total, 98 patients (62%) could be classified as LGE negative and 61 patients (38%) as LGE positive. LGE-positive patients were more often male (70% vs. 51%, *p* = 0.015) and showed a higher incidence of coronary artery disease at baseline (52% vs. 8% *p* < 0.001), but no differences in LVEF, cardiac biomarker levels or the severity of HF symptoms—determined by NYHA class—could be observed.

### 3.4. Diffuse Fibrosis of the Non-Infarcted Myocardium and Cardiac Biomarkers

Subsequently, LGE-positive patients were stratified based upon diffuse fibrosis of the non-LGE myocardium (above or below median post-contrast T1 time; the median T1 time was 382 ms, *n* = 24 in both groups). For the patient selection flow chart, see Figure 2. In both groups, the majority (*n* = 18, 75% in both) of patients showed a subendocardial or transmural LGE pattern, reflecting myocardial ischemia. Patients with shorter post-contrast T1 time showed higher levels of both NT-proBNP (1611 ng/L vs. 453 ng/L, *p* = 0.009) and galectin-3 (20.3 µg/L vs. 14.5 µg/L, *p* = 0.011) (Figure 3, Table 2), without differences in LVEF (33 [23–40] vs. 36 [30–45], p = 0.10) or cardiac dimensions (e.g., LVEDV and LVESV) between both groups.

### 3.5. Presence of Focal Fibrosis as A Determinant of Outcome

During a median follow-up of 2.6 [2.1–3.1] years, 10 (16%) patients experienced death and 10 (16%) were hospitalized for HF, compared to 12 (12%) and 11 (11%) of the LGE-negative patients. In total, 15 (25%) LGE-positive patients experienced the primary composite outcome compared to 20 (20%) of the LGE-negative patients. Using Kaplan–Meier analysis, no difference in event-free survival was observed between LGE-positive and LGE-negative patients (log-rank *p =* 0.69).

### 3.6. Presence of Diffuse Fibrosis of the Non-Infarcted Myocardium as A Determinant of Outcome

In LGE-positive patients, HF hospitalization and all-cause mortality were reached in 9 (43%) patients with a shorter post-contrast T1 time, compared to 3 (14%) patients with a longer T1 time during follow-up. Those patients with a shorter post-contrast T1 time had a worse prognosis regarding HF hospitalizations and all-cause mortality compared to those with a longer T1 time (log-rank *p* = 0.01, Figure 4), mainly due to higher HF hospitalization rates (log-rank *p* = 0.02) rather than death of any cause (log-rank *p* = 0.16). In univariable Cox regression analyses, patients with a shorter T1 time showed a higher risk of HF hospitalization and all-cause mortality (HR 1.09 [1.02–1.18] per 10 milliseconds decrease in T1 time, *p* = 0.013). Due to a small number of events, no multivariable regression analysis was performed.

As a sensitivity analysis, post-contrast T1 time was also determined in LGE-negative patients. LGE-negative patients were stratified according to the median of the post-contrast T1 time (median T1 time 405 [372–436] ms). In LGE-negative patients, hospitalization for HF and death occurred in 8 (18%) of the patients with a shorter post-contrast T1 time and in 11 (25%) of the patients with a longer post-contrast T1 time. No association with cardiac biomarker levels nor prognosis (log-rank *p* = 0.24) could be observed.

## 4. Discussion

Our results indicate that diffuse fibrosis (1) is evident in the non-infarcted area of the myocardium, (2) is associated with biomarkers of cardiac stretch and remodeling, and (3) has higher prognostic value regarding adverse outcomes compared to focal fibrosis in outpatient HF patients with evident myocardial scarring (Graphical abstract).

First, we show that diffuse fibrosis develops through the entire non-infarcted myocardium, alongside evident focal scarring, as a response to tissue damage after MI in mice. On the cardiac gene expression level, fibrosis is also evident, as depicted by a 5-fold increase in expression of galectin-3. Regarding outcome, diffuse fibrosis shows a strong association with cardiac function, as reflected by LVEF. Over the past decade, it is becoming increasingly clear that myocardial fibrosis plays a major role in the pathophysiology of CV disease [13]. Diffuse or interstitial fibrosis actively contributes to left ventricular dysfunction and is suggested to play a primary role in HF pathogenesis [14], which is consistent with our animal experiments. To validate the role of fibrosis in humans, diffuse and focal fibrosis were assessed using CMR. Plasma NT-proBNP and galectin-3 were significantly higher in patients with a shorter post-contrast T1 time, independent of LVEF or cardiac dimensions, in contrast to focal myocardial scarring. This indicates a clear association of interstitial fibrosis with cardiac wall stress and remodeling, and underscores the potential of galectin-3 as a marker of early cardiac remodeling, even before an evident HF phenotype arises [15,16,17]. From experimental studies, it is known that galectin-3 levels are at their top at peak fibrosis, but virtually absent after full recovery [18,19]. This might imply a significant role for galectin-3 in the active process of diffuse fibrogenesis [20], rather than focal scarring or replacement fibrosis.

LGE is known for its diagnostic accuracy in the assessment of focal scar formation and has high prognostic value, especially concerning sudden cardiac death and (life-threatening) ventricular arrhythmias [21,22]. However, in our study, no difference in outcome was observed between patients with LGE presence or absence. In (stable) outpatient HF patients with focal myocardial scarring, the critical period of scar formation may already have taken place, and has apparently not resulted in mortality due to arrhythmogenic episodes. This indicates the value of diffuse fibrosis in outcome assessment in outpatient HF patients.

In this study, diffuse myocardial fibrosis was assessed with a post-contrast T1 time of the non-infarcted myocardium and predicted prognosis. While 14% of patients with long T1 time were readmitted for HF or died of any cause, this concerned 43% of the patients with a short T1 time. This is in line with several studies emphasizing the prognostic importance of interstitial fibrosis in non-ischemic cardiomyopathy [23,24], and the non-infarcted myocardium [20,25] in coronary artery disease. A study by Marques and colleagues demonstrated that cardiac fibrogenesis even takes place and predicts prognosis in healthy individuals from the Multi-Ethnic Study of Atherosclerosis (MESA) study [26]. A high native T1 time (pre-contrast, >984 ms) was associated with a 2.1-fold increased relative risk of new-onset CV events, while high extracellular volume (ECV > 30%) showed a 2.0 fold, 2.9 fold and 1.7 fold increased risk for new-onset CV events, new-onset HF and all-cause mortality, respectively. This is in contrast to focal scarring by LGE, which did not show associations with outcome. This suggests a prominent role for pre-symptomatic diffuse fibrosis in early disease pathogenesis.

Previous human studies have shown the high accuracy of T1-based metrics (e.g., native T1, post-contrast T1 and ECV) in discriminating healthy from fibrotic cardiac tissue, and are associated with the presence and extent of diffuse fibrosis determined by cardiac biopsy and after autopsy [7]. However, to date, there is no consensus on the CMR mapping technique to be used. In some studies, investigators even developed a personalized T1 mapping approach to determine patient-specific focal and interstitial fibrosis [27].

Preclinical studies and clinical trials targeting cardiac fibrogenesis are emerging [28,29,30]. In the 1990s, The Randomized ALdactone Evaluation Study (RALES) confirmed the therapeutic protective effects of spironolactone in all-cause HF patients, which was most likely attributed to a reduction in myocardial fibrosis [31]. Additionally, in recent years, several studies have attempted to inhibit diffuse fibrosis, for instance, by targeting galectin-3 [29,30], a protein known to be actively involved in cardiac inflammation and remodeling. In a pre-clinical setting, galectin-3 inhibition results in a reduction in myocardial fibrosis and improvement of LV function [30]; however, to date, its favorable effects are difficult to demonstrate in patients [29]. Recently, the phase 2 Pirfenidone in Patients with Heart Failure and Preserved Left Ventricular Ejection Fraction (PIROUETTE) trial, showed the beneficial effects of pirfenidone—an FDA-approved drug for idiopathic pulmonary fibrosis—on CMR-assessed interstitial myocardial fibrosis [28].

Despite ample evidence on the beneficial effects of targeting diffuse fibrosis, translation to clinical practice has not been accomplished yet. The difficulty of determining a robust study endpoint is probably one of the major reasons, since this usually entails invasive processes (e.g., cardiac biopsy) or may not fully reflect cardiac remodeling (e.g., fibrotic biomarkers) [29]. Based on our study, in addition to already existing data [28], we believe that diffuse fibrosis and CMR metrics have the potential to be used as primary outcome parameters.

## 5. Conclusions

Together, our findings indicate that diffuse cardiac fibrosis plays an important role in active remodeling of the non-infarcted myocardium of patients with focal myocardial scarring. CMR-assessed diffuse fibrosis could provide an accessible and powerful noninvasive tool to determine prognosis in outpatient HF patients, and could potentially be used to determine the effectiveness of anti-fibrotic medication.

## 6. Limitations

A few limitations should be acknowledged. First, this study includes only a limited number of outpatient HF patients, preventing us from multivariable-adjusted analyses; as a result, conclusions should be drawn with some restraint. Second, the relationship of diffuse fibrosis with cardiac biomarkers and the prognostic value of this relationship regarding outcome were only observed in HF patients with focal myocardial scarring on CMR. In line with this, the majority of LGE-positive patients showed an ischemic LGE pattern. However, the LGE pattern of some patients was considered non-ischemic. The role of diffuse fibrosis and its usefulness regarding prognosis might differ between different HF etiologies, for instance, between ischemic and non-ischemic HF patients.

Because of the retrospective nature of this study, it was not possible to determine the relation and prognostic value of other (cardiac) biomarkers, for instance soluble ST2 and growth differentiation factor-15 (GDF-15), both known to be involved in inflammation and fibrosis.

Additionally, T1-based indices may be affected by other pathological processes (e.g., edema), and are highly dependent on CMR equipment and the time between contrast agent administration and measurements. Therefore, it is difficult to directly compare (post-contrast) T1 times between different studies and hospitals. Additionally, in this study, only post-contrast T1 time was determined, while ECV is increasingly being used to differentiate cardiomyopathy etiology. It would have been interesting to compare prognostic performance of different CMR metrics.

When using CMR as a robust study endpoint for diffuse fibrosis, it is worth considering the cost-effectiveness of CMR, especially in healthy subjects. Since the average price of CMR is EUR 160 to 1400 per scan [32], the costs may not outweigh the benefits in a relatively healthy population.

## Figures and Tables

**Figure 1 biomolecules-13-00410-f001:**
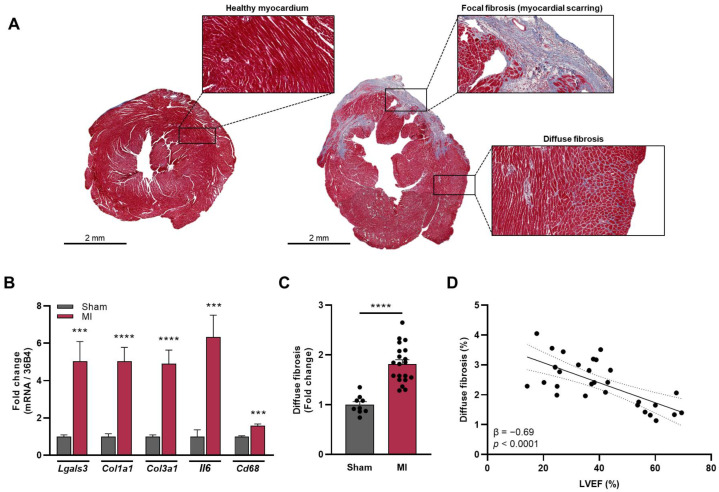
(**A**) Representative images of Masson’s trichrome-stained cardiac tissue sections of C57Bl/6J background mice 6 weeks post-MI, and their sham-operated counterparts. The image details represent a healthy myocardium, focal fibrosis and diffuse fibrosis. (**B**) Cardiac gene expression of genes involved in fibrosis and immune infiltration in sham-operated mice (*n* = 9) and mice 6 weeks post-MI (*n* = 20), depicted as fold change compared to sham. (**C**) Percentage of diffuse fibrosis of the non-infarcted myocardium in sham-operated mice (*n* = 9) and mice 6 weeks post-MI (*n* = 20), quantified from Masson’s trichrome-stained images and depicted as fold change compared to sham. (**D**) Association between diffuse fibrosis and cardiac function. *** *p* < 0.001, **** *p* < 0.0001. *Cd68*, cluster of differentiation 68; *Col1a1*, collagen type I alpha I chain; *Col3a1*, collagen type III alpha I chain; *Il6*, interleukin-6; *Lgals3*, galectin-3; MI, myocardial infarction.

**Figure 2 biomolecules-13-00410-f002:**
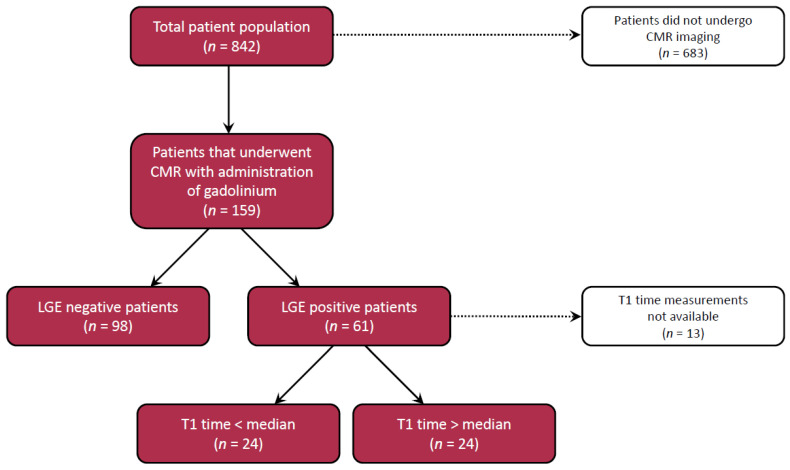
Patient selection flow chart.

**Figure 3 biomolecules-13-00410-f003:**
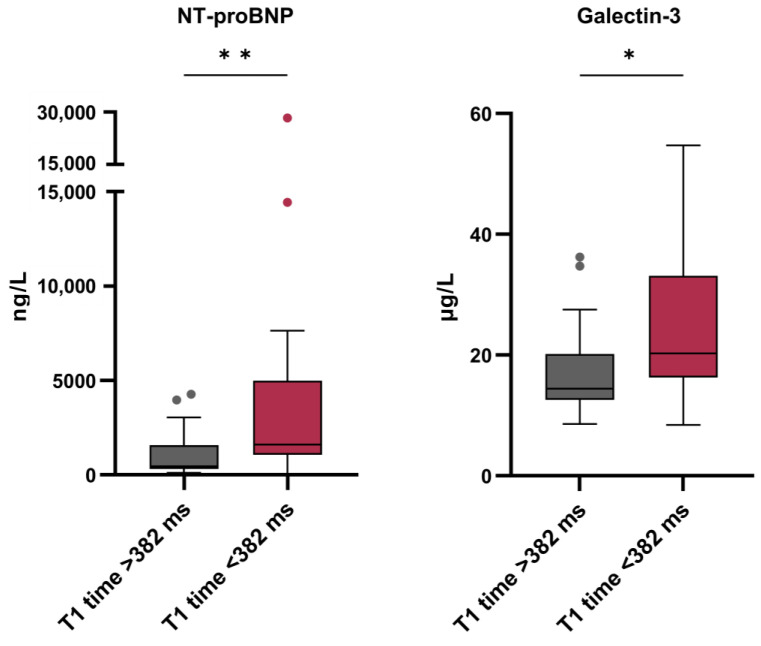
Biomarker levels of NT-proBNP and galectin-3 in LGE-positive patients stratified based upon median post-contrast T1 time of the non-infarcted myocardium. * *p* < 0.05, ** *p* < 0.01.

**Figure 4 biomolecules-13-00410-f004:**
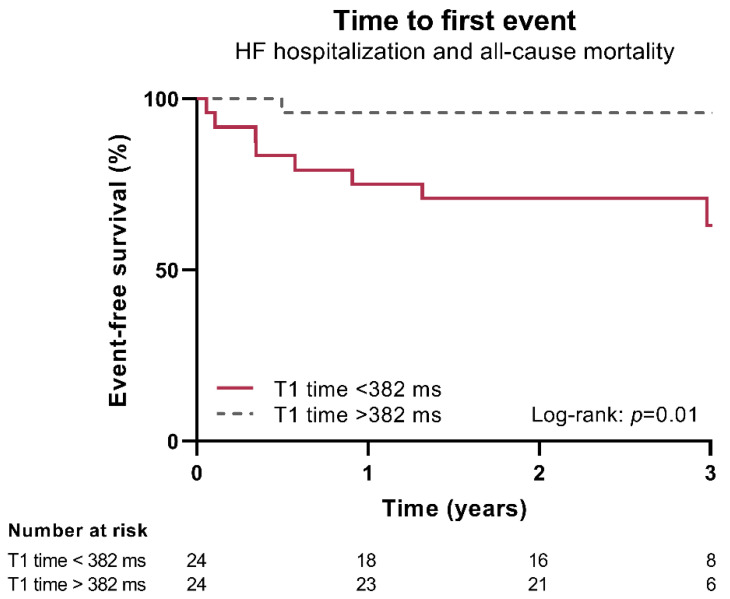
Kaplan-Meier survival curve for the primary combined endpoint (i.e., HF hospitalization and all-cause mortality) in LGE-positive patients, stratified by median post-contrast T1 time of the non-infarcted myocardium.

**Table 1 biomolecules-13-00410-t001:** Baseline characteristics of the total study cohort, divided by the presence of LGE.

Characteristics	Total Cohort *n* = 159	LGE Negative *n* = 98	LGE Positive *n* = 61	*p*-Value
Age (y), mean (SD)	59 (14)	58 (14)	60 (13)	0.40
Female sex, n (%)	66 (42)	48 (49)	18 (30)	0.015
SBP (mmHg), mean (SD)	120 (20)	122 (20)	116 (21)	0.075
DBP (mmHg), mean (SD)	73 (11)	74 (12)	71 (10)	0.081
BMI (kg/m2), mean (SD)	28 (5)	28 (6)	27 (4)	0.063
Heart failure history, n (%)				
Coronary artery disease	40 (25)	8 (8)	32 (52)	<0.001
Hypertension	64 (40)	40 (41)	24 (39)	0.85
Atrial fibrillation	42 (26)	29 (30)	13 (21)	0.25
NYHA class, n (%)				0.33
I	33 (21)	22 (22)	11 (18)	
II	90 (57)	51 (52)	39 (64)	
III	36 (22)	25 (26)	11 (18)	
Medication, n (%)				
β-blocker	145 (91)	88 (90)	57 (93)	0.43
ACEi/ARB	138 (87)	84 (86)	54 (89)	0.61
Diuretic	105 (66)	65 (66)	40 (66)	0.92
Aldosterone antagonist	73 (46)	37 (38)	36 (59)	0.009
Laboratory measurements				
eGFR (mL/min/1.73 m2), mean (SD)	72 (30)	72 (31)	71 (29)	0.78
NT-proBNP (ng/L), median [IQR]	753 [235–2137]	606 [172–1929]	1082 [381–2530]	0.052
Galectin-3 (μg/L), median [IQR]	16.6 [13.1–24.1]	16.7 [12.8–22.9]	16.6 [13.5–27.5]	0.49
CRP (mg/L), median [IQR]	4.0 [1.6–7.1]	4.3 [1.7–7.4]	3.1 [1.3–6.2]	0.42
Sodium (mmol/L), mean (SD)	140 (3)	141 (3)	139 (3)	0.016
CMR parameters				
LVEF (%), median [IQR]	38 [29–45]	37 [25–48]	35 [28–47]	0.77
Post-contrast T1 time (ms), median [IQR]	390 (62)	405 [372–436]	382 [347–426]	0.13
LVEDV (mL), mean (SD)	232 (77)	231 (76)	234 (80)	0.83
LVESV (mL), mean (SD)	152 (73)	150 (70)	156 (78)	0.59
LVSV (mL), mean (SD)	79 (28)	80 (29)	78 (26)	0.68

Abbreviations: ACEi, angiotensin converting enzyme inhibitor; ARB, angiotensin II receptor blocker; BMI, body mass index; CMR, cardiac magnetic resonance; CRP, C-reactive protein; DBP, diastolic blood pressure; eGFR, estimated glomerular filtration rate; LGE, late gadolinium enhancement; LVEDV, left ventricular end-diastolic volume; LVEF, left ventricular ejection fraction; LVESV, left ventricular end-systolic volume; LVSV, left ventricular stroke volume; NT-proBNP, N-terminal pro-B-type natriuretic peptide; NYHA, New York Heart Association; SBP, systolic blood pressure.

**Table 2 biomolecules-13-00410-t002:** Baseline characteristics of LGE-positive patients, divided by median post-contrast T1 time.

Characteristics	T1 Time < 382 ms *n* = 24	T1 Time > 382 ms *n* = 24	*p*-Value
Age (y), mean (SD)	62 (12)	56 (14)	0.11
Female sex, n (%)	11 (46)	5 (21)	0.066
Heart failure history, n (%)			
Coronary artery disease	11 (46)	10 (42)	0.77
Hypertension	9 (38)	8 (33)	0.76
Atrial fibrillation	5 (21)	4 (17)	0.71
NYHA class, n (%)			0.28
I	4 (17)	6 (25)	
II	14 (58)	16 (67)	
III	6 (25)	2 (8)	
Medication, n (%)			
β-blocker	22 (92)	22 (92)	1.00
ACEi/ARB	22 (92)	24 (100)	0.15
Diuretic	18 (75)	13 (54)	0.13
Aldosterone antagonist	15 (63)	14 (58)	0.77
Laboratory measurements			
NT-proBNP (ng/L), median [IQR]	1611 [1066–4406]	453 [323–1557]	0.009
Galectin-3 (μg/L), median [IQR]	20.3 [16.3–32.2]	14.5 [12.6–20.0]	0.011
Creatinine (μmol/L), median [IQR]	100 [84–135]	89 [74–99]	0.18
eGFR (mL/min/1.73 m2), mean (SD)	61 [41–79]	84 [62–98]	0.008
CMR parameters			
LVEF (%), median [IQR]	33 [22–36]	33 [23–45]	0.35
Post-contrast T1 time (ms), median [IQR]	347 [303–365]	426 [410–443]	<0.001
LVEDV (mL/m2), median [IQR]	135 [100–149]	132 [103–159]	0.51
LVESV (mL/m2), median [IQR]	92 [66–113]	93 [62–115]	0.85
LV mass (g/m2), median [IQR]	53 [44–78]	59 [53–77]	0.18

Abbreviations: ACEi, angiotensin converting enzyme inhibitor; ARB, angiotensin II receptor blocker; CMR, cardiac magnetic resonance; eGFR, estimated glomerular filtration rate; LVEDV, left ventricular end-diastolic volume; LVEF, left ventricular ejection fraction; LVESV, left ventricular end-systolic volume; NT-proBNP, N-terminal pro-B-type natriuretic peptide; NYHA, New York Heart Association.

## Data Availability

Data presented in this study are available on request from the corresponding author.

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
