# Peer review of "Diffuse Myocardial Fibrosis on Cardiac Magnetic Resonance Imaging Is Related to Galectin-3 and Predicts Outcome in Heart Failure"

_biomolecules, 2023, doi:10.3390/biom13030410_

Round 1

Reviewer 1 Report

The work performed by Sceever and cols. is valuable in its pre-clinical side and presents interesting data in the clinical one. However, some changes may be achieved to better understand the work, in my opinion.

-        METHODS

1.      I miss a description of the design with clear primary and secondary endpoints. Number of mice analysed should be shown. The paper is the addition of two different works, linked by their results but there should be a clear endpoint (combined or separated) to analyse the meaning of results. Please, add.

2.      Explain why post-contrast T1 is selected as diffuse myocardial fibrosis when gold standard is ECV. Explain why is has not been measured Native T1 and T2 that can confound the interstitial space, while it can be increased by other different proteins/molecules different from collagen and then, T1 post-contrast alone does not reflect specifically the fibrosis in the myocardium.

3.      Explain why diffuse fibrosis is not measured in none-LGE patients.

4.       Define pre-clinical and clinical events according to previous endpoints: What is diffuse fibrosis in the heart, what are normal values, etc… and what clinical outcomes are considered (heart failure admission or decompensation, cardiovascular death, non-cardiovascular death, sudden death, ICD-shocks…. Please, clarify.

5.      Explain why only Galectin-3 and NTproBNP are selected as biomarkers when other biomarkers of fibrosis could be used: ST2, collagen derived products….

-        RESULTS

1.      Please, further describe mice-related results, explain findings (type of fibrosis, amount of fibrosis, distribution of diffuse patterns, other pathologic findings as cellular component of white infiltrates, or macrophagues, etc…)

2.      Table 1. LVEF is measured by Echo or CMR? Set with CMR variables.

3.      Table 1 and 2: T1 relaxation time and post-contrast T1 time are the same? Unify.

-        DISCUSSION

1.      Discussion starts with the sentence “we demonstrate….” With a single-centre, not controlled, retrospective, univariable analysis and without expose clearly the results of preclinical study; which sounds a bit optimistic. Please modulate and extend preclinical results, as commented previously.

2.      Myocardial fibrosis out of dense scar has been previously described after acute myocardial infarction. Discuss what’s new with mice findings and its repercussion.

3.      Link and discuss what add the pre-clinical studies to clinical findings. It seems separate studies that has been artificially fixed because the relationship between them is not clearly explained.

4.      Discuss or add to limitations whether post-contrast T1 time is the way of measure diffuse fibrosis. It is important to conclude current “conclusion”.

Reviewer 2 Report

The authors aimed to study the presence and extent of diffuse fibrosis in the non- infarcted area of the myocardium (using a MI murine model). They also tested if the myocardial fibrosis (focal or diffuse) correlates with the heart failure biomarkers and with the prognosis in outpatient heart failure patients.

The topic is important and interesting. The manuscript is well written.

Some comments.

It should be stated how many mice were studied.

Some information about the patients' follow-up should be in the Methods section.

There is no Figure 1B

For T1 time analysis it should be clearly stated in the text how many patients were studied – in the text there is n=24 (line 183), but then form table 2 we learn that it was 24 in each group. How were the patients chosen and why 13 patients were excluded (61 minus 48)?

The term ECV (line 255) is not explained.

The T1 time had prognostic value only in patients with focal fibrosis. This should be stated in the conclusions/limitations – the results refer only to the patients with focal fibrosis (if I understand well lines 216-218). From the study we can learn that it is about 40% of the whole cohort of outpatient heart failure patients. In the rest of HF patients it doesn’t play a role.

There is one more important limitation of the study – there is a mix of patients with ischemic and non-ischemic heart failure. The distribution of fibrosis and the role of fibrosis in these groups may differ.

Round 2

Reviewer 1 Report

I would like to congratulate authors by their answers to revision. Now the work looks more understandable and rigorous. 

Author Response

We thank the reviewer for their final comments that helped our manuscript substantially improve.